# Clinical Value of EZH2 in Hepatocellular Carcinoma and Its Potential for Target Therapy

**DOI:** 10.3390/medicina58020155

**Published:** 2022-01-20

**Authors:** An-Na Bae, Soo-Jung Jung, Jae-Ho Lee, Hyunsu Lee, Seung Gyu Park

**Affiliations:** 1Department of Anatomy, Keimyung University School of Medcine, Daegu 42601, Korea; anna@dsmc.or.kr (A.-N.B.); soojung@naver.com (S.-J.J.); anato82@dsmc.or.kr (J.-H.L.); neuroana@dsmc.or.kr (H.L.); 2Department of Radiation Oncology, Keimyung University Dongsan Medical Center, Keimyung University School of Medicine, Daegu 42601, Korea

**Keywords:** hepatocellular carcinoma, EZH2, STAT3, TCGA

## Abstract

*Background and objectives:* EZH2 is overexpressed in hepatocellular carcinoma (HCC) and is correlated with poor prognosis. However, its clinical significance and molecular mechanism have not been studied in HCC. In this study, clinical and prognostic values of EZH2 was studied using Total Cancer Genome Atlas (TCGA) data and then, these data were confirmed in Huh1 and HepG2 cell lines. *Materials and Methods*: We used the TCGA database from cBioPortal. In addition, we analyzed EZH2 mRNA levels in HCC cell lines and its correlation with STAT3 and EZH2. *Results:* According to TCGA, EZH2 had a prognostic value in various cancers, especially in HCC. Furthermore, EZH2 in HCC was correlated with N stage (*p* = 0.045) and alpha-fetoprotein (AFP) > 20 ng/mL (*p* < 0.01). However, a negative association between EZH2 and age (*p* = 0.027) was found. The overall survival result of HCC was significantly poorer in patients with high EZH2 expression. In addition, the recurrence rate was also significantly higher in patients with high expression of EZH2 than those with low expression (χ2 = 16.10, *p* < 0.001). EZH2 expression was negatively correlated with STAT3 expression among EZH2-associated genes (R = −0.163, *p* = 0.002). EZH2 expression level was down-regulated to 50% or less compared to the control group treated negative siRNA. MTT assays showed that EZH2-siRNA affected on the viability of HCC cell line significantly. *Conclusions:* In conclusion, the overexpression of EZH2 was an independent biomarker for poor outcomes of HCC. However, more in vivo studies are required to identify the downstream target genes in HCC to improve our understanding of the biological role of EZH2 in HCC.

## 1. Introduction

Hepatocellular carcinoma (HCC) is one of the leading life-threatening malignancies and the second most common cause of cancer-related deaths globally [1]. The global incidence of HCC has been increasing, with estimated 600,000–800,000 new cases occurring annually [2]. The development of surgical techniques has improved the prognosis of patients with HCC [3]. However, the prognosis of HCC remains dismal despite the great advances in HCC treatment. Thus, the need for reliable biomarkers for diagnosing HCC and novel strategies for the effective treatment of patients with HCC is pressing.

HCC-associated genes, along with a subset of their neighbors, are generally developed into a gene interaction network [4]. Research has shown that various molecules play key roles in the development and progression of HCC: For example, there are genes such as signal transducer and activator of transcription 3 (STAT3) and centrosomal protein of 55 (CEP55) [1,5]. In HCC, STAT3 and CEP55 are known to be involved in cell migration and invasion [2].

Enhancer of zeste homolog 2 (EZH2) is the catalytic subunit of polycomb repressive complex 2, and it has been functionally associated with the regulation of the cell cycle [6]. As EZH2 regulates cell cycle progression, its dysregulation accelerates cell proliferation and prolongs cell survival, which may lead to carcinogenesis and cancer development [7]. Recently, several studies have shown that EZH2 is aberrantly upregulated in various malignant tumors, such as prostate and breast cancer, and is associated with advanced stages and poor prognosis [6]. Especially, EZH2 is overexpressed in HCC, and this correlates with poor prognosis [6]. However, its clinical significance and molecular mechanisms have not been elucidated to date.

In this study, we aimed to analyze the clinical and prognostic values of EZH2 expression in hepatocellular carcinomas using the Total Cancer Genome Atlas (TCGA) data. We also analyzed the association between EZH2 and STAT3. Furthermore, based on these data, cell invasion and migration in HCC cell lines transfected with EZH2 siRNA were investigated to clarify the precise underlying mechanism of HCC. The results of this study potentially reveal new targets and strategies that can be applied to HCC diagnosis and treatment.

## 2. Materials and Methods

### 2.1. The Cancer Genome Atlas (TCGA) Data Analysis

We investigated the publicly available TCGA datasets, and the relevant data were downloaded from the TCGA Data Portal [8]. The microarray and RNA-Seq experiments and clinical data were downloaded directly from the TCGA website in March 2021. The TNM stage was evaluated according to the seventh edition of the American Joint Committee on Cancer staging system.

### 2.2. Cell Culture and siRNA Transfection

The Huh1 and HepG2 cell lines were obtained from Dr. Yun-Han Lee. He obtained Huh1 and HepG2 from the Japanese Collection of Research Biosources Cell Bank (JCRB) and American type culture collection (ATCC), respectively. HepG2 and Huh1 cells were cultured in Dulbecco’s modified Eagle medium supplemented with 1% penicillin/streptomycin solution and 10% fetal bovine serum (FBS) (Gibco BRL., Grand Island, NY, USA) in a humidified 5% CO_2_ incubator at 37 °C. The targeting sequence of EZH2-siRNA was as follows: 5-AACCATGTTTACAACTATCAA-3 (Qiagen, Cat.no. SI02665166). Lipofectamine (Invitrogen) and 200 nM siRNAs were mixed in Opti-MEM (Thermo Scientific, Rockford, IL, USA). After 48 h of EZH2-siRNA transfection, the RNA was extracted. The total cellular RNA was extracted using a QIAzol lysis reagent (Qiagen, Redwood City, CA, USA) according to the manufacturer’s protocol. The NanoDrop ND-1000 spectrophotometer (Thermo Fisher Scientific, Waltham, MA, USA) was used to determine the quantity and quality of the isolated total cellular RNA.

### 2.3. Quantitative Real-Time PCR Analysis (RT-qPCR)

Reverse-transcription reactions were conducted using the ReverTra Ace qPCR RT Master Mix (TOYOBO, Osaka, Japan). The expression levels of EZH2, STAT3, and GAPDH were measured by RT-qPCR. RT-qPCR analysis was performed using a CFX Connect RT-PCR System (Bio-Rad, Hercules, CA, USA). The primers used for RT-qPCR were synthesized by Bionics (Seoul, Korea). The primer sequences used for RT-qPCR are listed in Table 1. The PCR amplification cycles were as follows: 95 °C for 10 min, followed by 40 cycles of 95 °C for 60 s and 72 °C for 30 s.

### 2.4. Cell Viability Assay

HepG2 and Huh1 cells were seeded into 24-well plates at a density of 1 × 10^5^ cells/well. After 24 h, 48 h, and 72 h after siRNA transfection at 37 °C, the cells were subsequently incubated with 100 μL of 5 mg/mL MTT for 4 h. Cell viability was subsequently analyzed at a wavelength of 570 nm using an Asys UVM 340 microplate reader (Biochrom, Cambridge, UK). Each experiment was performed in triplicate.

### 2.5. Wound Healing Assay

HepG2 and Huh1(1 × 10^5^) were seeded in 12-well plates at 70–80% confluence for the wound healing assay. After washing, the cells were transfected with 200 nM negative control siRNA or EZH2 siRNAs using Lipofectamine 2000 (Invitrogen, Camarillo, CA, USA) according to the manufacturer’s instructions. After 6 h of transfection, the medium was replaced with a standard culture medium. The cells were incubated at 37 °C for 24 h, 48 h, and 72 h. Light microscope images of three locations of the marked wounds were obtained, and the migrated cells were counted.

### 2.6. Statistical Analysis

For statistical analyses, the Statistical Package for the Social Sciences (SPSS), (v24, IBM, Armonk, NY, USA) was used. The Chi-squared test, Fisher’s exact test, Mann–Whitney U-test, and simple correlation tests were used to analyze the association between variables. Univariate survival analysis was performed using the log-rank test with Kaplan–Meier curves. Overall survival was defined as the time between diagnosis and mortality. Recurrence-free survival was defined as the time between diagnosis and disease recurrence or the development of distant metastasis. Hazard ratios (HR) and 95% confidence intervals (CI) were estimated using the multivariate Cox proportional hazard model, with adjustment for age, T stage, N stage, M stage, APF level, and EZH2 expression. Statistical significance was set at *p* < 0.05.

## 3. Results

### 3.1. Clinical Characteristics of EZH2 Expression in Hepatocellular Carcinoma

Based on the TCGA data, EZH2 expression was associated with the prognoses for various cancers, and this was most significant for HCC. Stratified by survival, HCC ranked 36th, with statistical significance of the prognostic value of EZH2 in cancer types at *p* < 0.001 (Table 2). The clinical data for HCC were obtained from TCGA [8], and they included age, gender, grade, clinical stage, and TNM stage of 360 HCC patients. To identify clinical value of EZH2, patients were categorized into two subgroups, according to the median value of EZH2 expression. The association between age, sex, T stage, N stage, M stage, AFP and Child–Pugh class is presented in Table 3. The results showed that EZH2 expression was positively correlated with age *(p* = 0.027), N stage (*p* = 0.045), and alpha-fetoprotein (AFP) >20 ng/mL (*p* < 0.01) (Table 3). The EZH2 expression levels were higher in patients younger than 65 years than in those older. Regarding AFP, which is used as a marker for HCC, the level of EZH2 expression was higher in patients with concentrations of >20 ng/mL than of <20 ng/mL. In addition, the level of expression of EZH2 at N1 was higher than that at N0. The other clinical features showed no significant differences.

### 3.2. Prognostic Value of EZH2 mRNA Expressions in HCC

Survival analysis of HCC was performed to determine the prognostic value of the EZH2 mRNA expression. When the association between EZH2 expression and overall survival of HCC patients was investigated, the Kaplan–Meier analysis found that high expression of EZH2 was correlated with significantly poorer prognoses of patients with high EZH2 expression (χ2 = 16.10, *p* < 0.001) (Figure 1A). In addition, the recurrence rate was also significantly higher in patients with high EZH2 expression than in those with low expression (χ2 = 12.70, *p* < 0.001) (Figure 1B). High EZH2 expression was associated with a poor prognosis. Multivariate analysis showed that EZH2 expression (hazard ratio [HR], 1.92; 95% confidence interval [CI], 1.073–3.425, *p* = 0.028) and age (HR, 1.034; 95% CI, 1.007–1.062, *p* = 0.015) were significant independent prognostic factors for overall survival. However, they did not have statistical significance for recurrence rate.

### 3.3. Correlation with Downstream Genes Related to EZH2

Quantitative correlation analysis was performed using clinical parameters. The gene-gene correlation analysis based on the TCGA data analysis showed that EZH2 expression was negatively correlated with STAT3 expression (R = −0.163, *p* = 0.002) (Figure 2A). We also examined the association between EZH2 and the CEP55 gene, a subgene of EZH2. The levels of expression of EZH2 and CEP55 were positively correlated (R = 0.601, *p* < 0.001) (Figure 2B).

### 3.4. Effect of EZH2 Silencing on STAT3 mRNA Expression

To investigate the association between EZH2 and STAT3, mRNA expression of STAT3 was analyzed in EZH2-silencing HepG2 and Huh1 cells. EZH2 was knocked down to determine its association with STAT3 expression. The HepG2 and Huh1 cells were transfected with NC siRNA and EZH2-siRNA. To evaluate the potential role of EZH2 in regulating STAT3, we first investigated the effect of its expression on the efficacy of EZH2-siRNA by qPCR. As shown in Figure 3, EZH2 mRNA was significantly lower in HepG2 and Huh1 cells transfected with EZH2-siRNA than in cells transfected with NC-siRNA. In addition, EZH2 silencing induces a decrease in mRNA levels of STAT3 in both cells. These results suggest that the function of EZH2 is related to the STAT3 pathway in HCC progression.

### 3.5. EZH2 Knockdown Reduces HCC Cell Viability and Cell Recovery

The viability of the HepG2 and Huh1 cells was assessed using an MTT assay. EZH2 knockdown significantly decreased HepG2 and Huh1 cell viability after 24 h, 48 h, and 72 h (Figure 4).

These findings suggest that EZH2 regulates HCC cell viability. In addition, wound healing analysis was used to determine the extent of resilience after treatment with EZH2-siRNA knockdown (Figure 5). The resilience in HepG2 was significantly decreased (Figure 5A), but the resilience in Huh1 did not show any significant results (Figure 5B).

## 4. Discussion

In this study, the clinical characteristics of HCC were confirmed using TCGA and the function of EZH2 was investigated using HCC cell line. As a novel and potential target for cancer therapy, EZH2 has become extensively researched [7]. Several new drugs targeting EZH2 are being developed and evaluated in clinical trials. Therefore, targeting the carcinogenic activity of EZH2 in HCC may improve diagnosis and prognosis.

The analysis of clinical features showed that it was related to the degree of metastasis of lymph nodes, and it was also significantly related to an increase in the concentration of AFP. Previous studies have shown that AFP promotes invasion and metastasis in HCC [1,3,4]. AFP is a representative biomarker for HCC, suggesting that elevated serum AFP concentrations (>20 ng/mL) correlate with an increase in the risk of HCC development [9]. Given this result, EZH2 expression seems to affect AFP elevation, inducing poorer prognosis. Although the exact mechanism is not known yet, the expression of EZH2 is thought to affect lymph node metastasis and deteriorate the prognosis of HCC. According to previous research, EZH2 upregulation was associated with HCC progression and multiple HCC metastatic features, including venous invasion, direct liver invasion, and the absence of tumor encapsulation [10]. In addition, the results of previous studies on the correlation between EZH2 and lymph node metastasis showed that the correlation was stronger for EZH2-expressing tumor cells in lymph nodes than for matched primary tumor cells [11]. Therefore, high expression levels of EZH2 were associated with the pathological grade of tumors and lymph node metastasis.

EZH2 has downstream genes involved in signalling, and EZH2 interacts with these genes to cause carcinogenesis. Among related genes, it has been previously reported that overexpression of CEP55 may worsen the prognosis in HCC [12,13]. When looking at the association between CEP55 and EZH2 gene, there is a positive correlation. The relationship between the two genes has been reported in lung, but there is no study in HCC [14]. Moreover, STAT3 is known to cause carcinogenesis by binding to EZH2 and being activated [13]. EZH2 binds to and methylates STAT3, leading to the enhancement of its activity by increasing its tyrosine phosphorylation. STAT3 and EZH2 are potential molecular biomarkers for tumor progression and serve as poor predictors of outcomes. Previous research has suggested that STAT3 and EZH2 are closely associated with cell proliferation, invasion, and metastasis [15]. The activation of EZH2 and STAT3 is significantly correlated with TNM stage and patient survival, suggesting that a combination of STAT3 and EZH2 expressions may determine the clinical TNM stage and predict disease outcomes.

We wanted to know why EZH2 and STAT3, which were negatively correlated in big data, were different in vitro. The HCC cell lines, HepG2 and Huh1 cells, were transfected with siRNA. Subsequently, their migration ability was determined using a wound-healing assay and their invasion ability was determined using the transwell assay. The results showed that the knockdown of EZH2 markedly decreased the migration and invasion abilities of the HepG2. However, when observed visually in Huh1, the cell mobility and recovery power seemed to decrease, but quantification did not show any significant results. More research is needed in this regard.

STAT3, which is closely related to cancer cell invasion and metastasis, was downregulated by EZH2 knockdown [7]. We study suggests that the downregulation of EZH2 and STAT3 can be utilized as new therapeutic target candidates for HCC. The present study demonstrates that EZH2 regulates cell viability, mobility, and resilience in HCC cell lines, which suggests that EZH2 can be used as a potential biomarker for HCC diagnosis and prognostication. The downregulation of EZH2 promotes apoptosis through suppression. Similarly, resilience was reduced in HCC. These in vitro results indicate that EZH2 may play an important role in the development and progression of HCC.

## 5. Conclusions

The overexpression of EZH2 was an independent biomarker for poor outcomes of HCC. Based on the results, EZH2 may be used as a therapeutic target in patients with HCC. However, more in vivo studies are required to identify the downstream target genes in HCC to improve our understanding of the biological role of EZH2 in HCC.

## Figures and Tables

**Figure 1 medicina-58-00155-f001:**
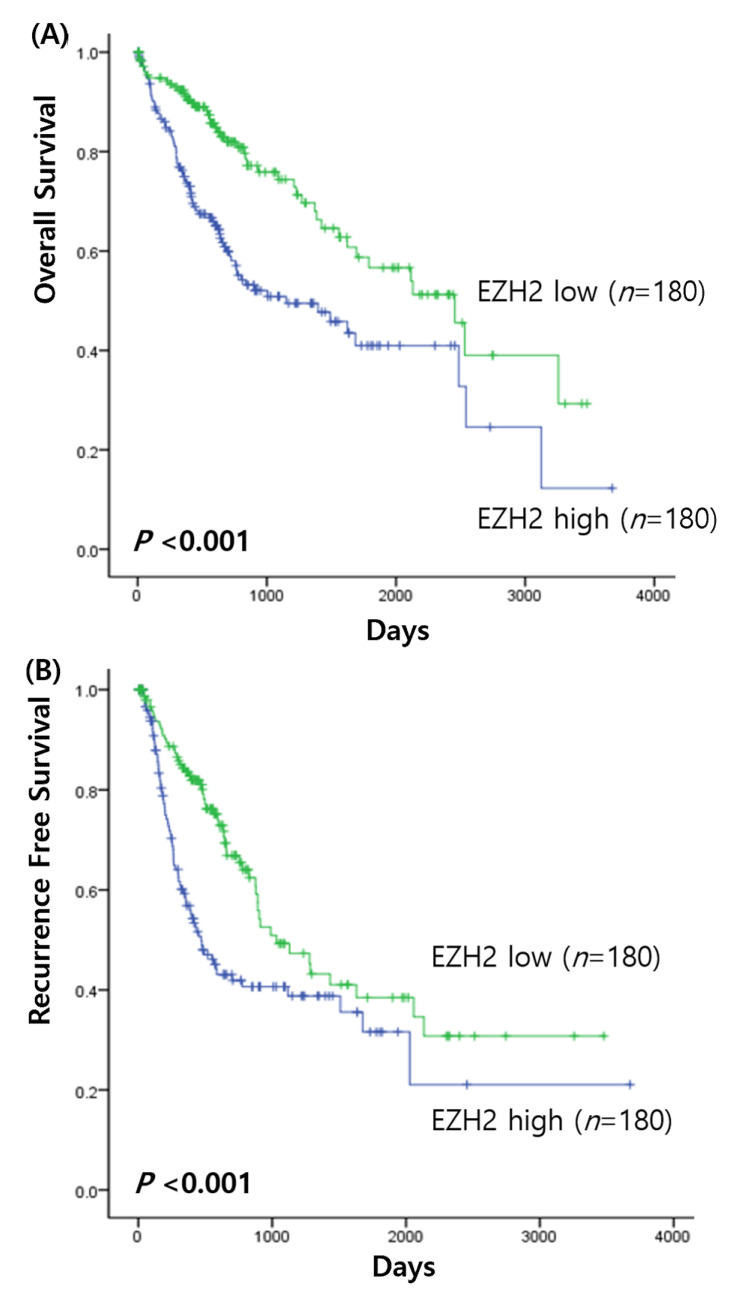
Survival analysis in HCC. (**A**) Overall survival of EZH2 expression; (**B**) recurrence free survival of EZH2 expression.

**Figure 2 medicina-58-00155-f002:**
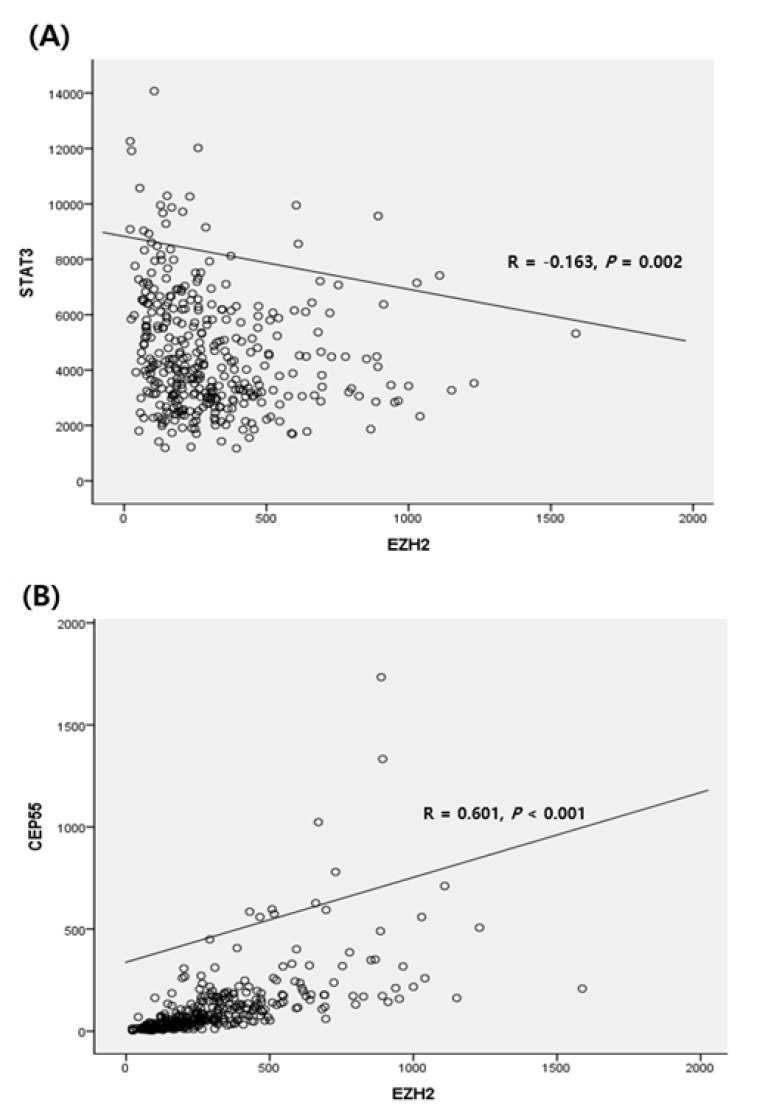
Correlation analysis: (**A**) between EZH2 STAT3 and (**B**) between EZH2 expression and CEP55.

**Figure 3 medicina-58-00155-f003:**
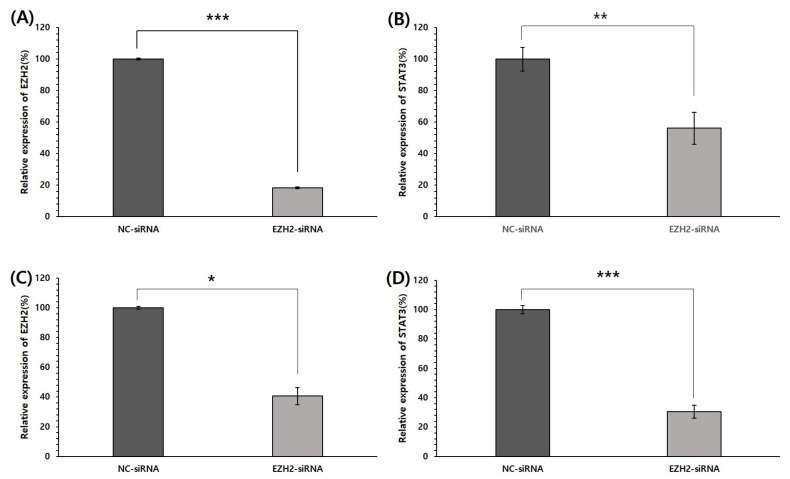
siRNA knockdown of EZH2 in HCC cells and the association between EZH2 and STAT3 mRNA expression. (**A**) Detection of EZH2 mRNA expression in HepG2 cells at 48 h after transfection. (**B**) After EZH2 knockdown, STAT3 mRNA expression in HepG2. (**C**) Detection of EZH2 mRNA expression in Huh1 cells at 48 h after transfection. (**D**) After EZH2 knockdown, STAT3 mRNA expression in Huh1. * *p* < 0.05, ** *p* < 0.01, and *** *p* < 0.001 by Student’s *t*-test.

**Figure 4 medicina-58-00155-f004:**
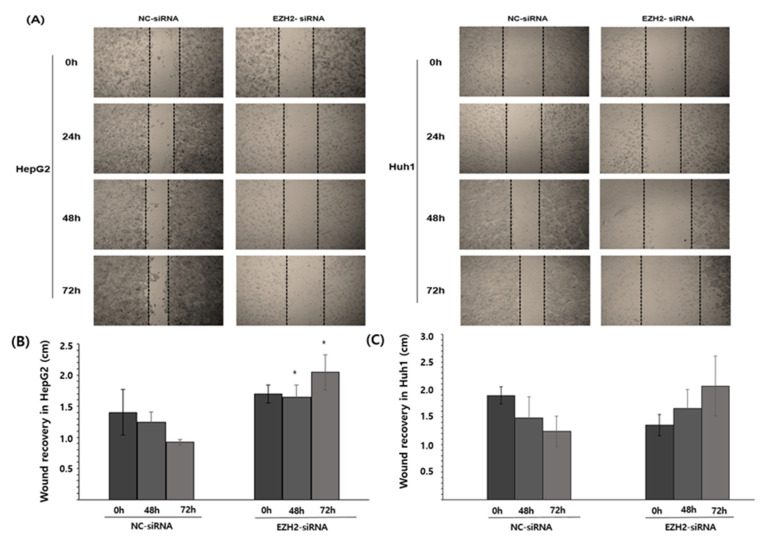
Cells migration in cell scratch wound healing assay. EZH2 inhibited cell migration in HepG2. (**A**) Wound healing assay of HepG2 and Huh1 upon (200 nM) for 24 h, 48 h, and 72 h. The HCCs treated with NC-siRNA were used as control. (**B**) Quantification of wound area in control and EZH2 treated HepG2. (**C**) Quantification of wound area in control and EZH2 treated Huh1. * *p* < 0.05.

**Figure 5 medicina-58-00155-f005:**
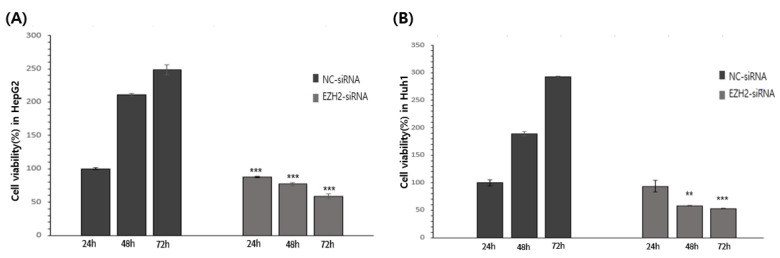
siRNA knockdown of EZH2 and growth of HCC cells. (**A**) Detection of cell viability after transfection in HepG2. (**B**) Detection of cell viability after transfection in Huh1. ** *p* < 0.01, and *** *p* < 0.001 by Student’s *t*-test.

**Table 1 medicina-58-00155-t001:** Primer sequences used for RT-qPCR in this study.

Name	Primer (5′ to 3′)
EZH2	Forward: GACCTCTGTCTTACTTGTGGAGC
	Reverse: CGTCAGATGGTGCCAGCAATAG
STAT3	Forward: GCTTTTGTCAGCGATGGAGT
	Reverse: ATTTGTTGACGGGTCTGAAGTT
GAPDH	Forward: GAAAGGTGAAGGTCGGAGTC
	Reverse: GTTGAGGTCAATGAAGGGGTC

**Table 2 medicina-58-00155-t002:** Statistical significance of prognostic value of EZH2 in various cancer types.

Cancer Type	*p*
Bladder urothelial carcinoma	0.726
Breast invasive carcinoma	0.402
Cervical squamous cell carcinoma	0.293
Colon adenocarcinoma	0.333
Esophageal carcinoma	0.837
Glioblastoma multiforme	0.994
Head and neck squamous cell carcinoma	0.183
Kidney renal clear cell carcinoma	<0.001 *
Kidney renal papillary cell carcinoma	0.00794 *
Acute myeloid leukemia	0.378
Brain lower grade glioma	0.00212 *
Liver hepatocellular carcinoma	<0.001 *
Lung adenocarcinoma	0.476
Lung squamous cell carcinoma	0.0946
Ovarian serous cystadenocarcinoma	0.526
Pancreatic adenocarcinoma	0.145
Rectum adenocarcinoma	0.959
Sarcoma	0.415
Skin cutaneous melanoma	0.182
Stomach adenocarcinoma	0.0874
Uterine corpus endometrial carcinoma	0.144

* *p* < 0.05.

**Table 3 medicina-58-00155-t003:** Clinical characteristics of EZH2 expression in hepatocellular carcinoma.

	EZH2 Expression		
	High (%, *n*)	Low (%, *n*)	*p* Value
Age			0.027 *
<65	56.0 (98)	44.0 (77)	
≥65	44.3 (82)	55.7 (103)	
Sex			0.176
Male	47.5 (116)	52.5 (128)	
Female	55.2 (64)	44.8 (52)	
T stage			0.386
T1	51.4 (90)	48.6 (85)	
T2	43.0 (40)	57.0 (53)	
T3	56.0 (42)	44.0 (33)	
T4	50 (7)	50 (7)	
N stage			0.045 *
N0	49.4 (117)	50.6 (120)	
N1	100 (4)	0 (0)	
M stage			0.614
M0	51.4 (129)	48.6 (122)	
M1	40.0 (2)	60.0 (3)	
AFP			<0.001 *
<20 ng/mL	34.9 (51)	65.1 (95)	
≥20 ng/mL	65.9 (85)	34.1 (44)	
Child–Pugh class			0.543
A	44.9 (97)	55.1 (119)	
B	45.0 (9)	55.0 (11)	
C	100 (1)	0 (0)	

* *p* < 0.05.

## Data Availability

The data presented in this study are available on request from the corresponding author.

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
