# Peer review of "Clinical Value of EZH2 in Hepatocellular Carcinoma and Its Potential for Target Therapy"

_medicina, 2022, doi:10.3390/medicina58020155_

Round 1
Reviewer 1 Report
This is a combined research reports using the established database and cultured HCC cells, which emphasizes an association between enhanced EZH2 expression and poor prognoses of HCCs. The authors adopted firm methodologies and their results seem to be reliable. However, since there are already many similar reports, the present investigation is insufficient to provide a meaningful progression to this research field. If the authors would like to resubmit this paper, I think that multivariate analyses are necessary to prove that EZH2 overexpression is not a cofounding factor in poor prognoses of HCCs.
Author Response
Dear Reviewer,
First of all we would like to thank the referees and editor for their time in reviewing our manuscript. Their constructive comments helped us to improve our work.
---------------------------------------------------------------------------------
This is a combined research reports using the established database and cultured HCC cells, which emphasizes an association between enhanced EZH2 expression and poor prognoses of HCCs. The authors adopted firm methodologies and their results seem to be reliable. However, since there are already many similar reports, the present investigation is insufficient to provide a meaningful progression to this research field. If the authors would like to resubmit this paper, I think that multivariate analyses are necessary to prove that EZH2 overexpression is not a cofounding factor in poor prognoses of HCCs.
--> multivariate analysis was performed and its data was added in 3.2 section.
Reviewer 2 Report
There is an interesting work about biomarkers in HCC, the manuscript is well redacted and comprehensive. But still, there are some things that need to be improved:
Methodologies
Please inform amplicon and TM of the primers.
Results:
The authors should explain the N, T, and M stages.
In table 3, put an asterisk in the significant comparisons
3.3 the authors should reinforce the data with a table (or graph?)
There is an error in the number of figures, figure 3 must say figure 2.
Line 137. The pronoun “I” is out of place
In figure 3, please enter in each chart the cell type analyzed (HepG2 or Huh1)
In Figure 4, graphics B and C are of poor quality, please improve them.
Discuss
Line 180-184 the authors should be more extensive in explaining the relation with AFP.
Line 200-206, why could STAT3 at cellular levels be a predictor of HCC prognosis? the authors may discuss these in-depth.
Line 207, is there literature on the relationship pf EZH2 and STAT3 in an animal models (not in vitro)?
Author Response
Dear Reviewer,
First of all we would like to thank the referees and editor for their time in reviewing our manuscript. Their constructive comments helped us to improve our work.
---------------------------------------------------------------------------------
There is an interesting work about biomarkers in HCC, the manuscript is well redacted and comprehensive. But still, there are some things that need to be improved:
Methodologies
Please inform amplicon and TM of the primers.
--> It was added in section 2.3.
Results:
The authors should explain the N, T, and M stages.
-->It was added in section 2.1.
In table 3, put an asterisk in the significant comparisons
-->It was added in Table 3.
3.3 the authors should reinforce the data with a table (or graph?)
There is an error in the number of figures, figure 3 must say figure 2.
-->It was revised.
Line 137. The pronoun “I” is out of place
-->It was revised.
In figure 3, please enter in each chart the cell type analyzed (HepG2 or Huh1)
-->It was revised.
In Figure 4, graphics B and C are of poor quality, please improve them.
-->It was added.
Discuss
Line 180-184 the authors should be more extensive in explaining the relation with AFP.
-->It was added.
Line 200-206, why could STAT3 at cellular levels be a predictor of HCC prognosis? the authors may discuss these in-depth.
-->It was added (references 13-15).
Line 207, is there literature on the relationship pf EZH2 and STAT3 in an animal models (not in vitro)?
--> Only molecular study was conducted, therefore, we studied its association in HCC patients data from TCGA.
Reviewer 3 Report
The article is sufficiently well written and the English language is fluent. The main finding was that the overall survival result of HCC was poorer in patients with high EZH2 expression. However I would suggest some minor revision to the manuscript:
- Please clearly specify the meaning of High and Low in relation to the expression of EXH2.
- It is difficult to understand the dimension of the statistical sample extracted from the TCGA dataset (no. of patients selected)
- In the method section it is not specified what test was used for the correlation analysis in part 3.3. I suggest to the authors to specify what test was performed in the statistical analysis section.
- I personally suggest to avoid the use of first-person (ex. line 137).
Author Response
Dear Reviewer,
First of all we would like to thank the referees and editor for their time in reviewing our manuscript. Their constructive comments helped us to improve our work.
---------------------------------------------------------------------------------
The article is sufficiently well written and the English language is fluent. The main finding was that the overall survival result of HCC was poorer in patients with high EZH2 expression. However I would suggest some minor revision to the manuscript:
- Please clearly specify the meaning of High and Low in relation to the expression of EXH2.
-->Patients were divided according to the median value of EZH2 expression. It was added in section 3.1.
- It is difficult to understand the dimension of the statistical sample extracted from the TCGA dataset (no. of patients selected)
--> In TCGA, HCC data has total 360 patients. We used them.
- In the method section it is not specified what test was used for the correlation analysis in part 3.3. I suggest to the authors to specify what test was performed in the statistical analysis section.
--> It was added in section 2.6.
- I personally suggest to avoid the use of first-person (ex. line 137).
--> It was revised.
Round 2
Reviewer 1 Report
The authors proved with multivariate analysis that EZH2 expression is an independent factor in patients' prognoses. The additional findings substantially raised scientific value of this article. It may contribute as one of data to support the current concept about EZH2 expression in HCCs.